# Centrosome Dynamics and Its Role in Inflammatory Response and Metastatic Process

**DOI:** 10.3390/biom11050629

**Published:** 2021-04-23

**Authors:** Massimo Pancione, Luigi Cerulo, Andrea Remo, Guido Giordano, Álvaro Gutierrez-Uzquiza, Paloma Bragado, Almudena Porras

**Affiliations:** 1Department of Sciences and Technologies, University of Sannio, 82100 Benevento, Italy; luigi.cerulo@unisannio.it; 2Pathology Unit, Mater Salutis Hospital AULSS9, “Scaligera”, 37122 Verona, Italy; alguuz@ucm.es; 3Department of Medical Oncology Unit, University of Foggia, 71122 Foggia, Italy; giordano.guido81@gmail.com; 4Department of Biochemistry and Molecular Biology, Faculty of Pharmacy, Complutense University Madrid, 28040 Madrid, Spain; andrea.remo@aulss9.veneto.it (Á.G.-U.); pbragado@ucm.es (P.B.); maporras@ucm.es (A.P.); 5Health Research Institute of the Hospital Clínico San Carlos (IdISSC), 28040 Madrid, Spain

**Keywords:** centrosome, chromosome instability, Rho GTPases, p38 MAPK, tumor microenvironment

## Abstract

Metastasis is a process by which cancer cells escape from the location of the primary tumor invading normal tissues at distant organs. Chromosomal instability (CIN) is a hallmark of human cancer, associated with metastasis and therapeutic resistance. The centrosome plays a major role in organizing the microtubule cytoskeleton in animal cells regulating cellular architecture and cell division. Loss of centrosome integrity activates the p38-p53-p21 pathway, which results in cell-cycle arrest or senescence and acts as a cell-cycle checkpoint pathway. Structural and numerical centrosome abnormalities can lead to aneuploidy and CIN. New findings derived from studies on cancer and rare genetic disorders suggest that centrosome dysfunction alters the cellular microenvironment through Rho GTPases, p38, and JNK (c-Jun N-terminal Kinase)-dependent signaling in a way that is favorable for pro-invasive secretory phenotypes and aneuploidy tolerance. We here review recent data on how centrosomes act as complex molecular platforms for Rho GTPases and p38 MAPK (Mitogen activated kinase) signaling at the crossroads of CIN, cytoskeleton remodeling, and immune evasion via both cell-autonomous and non-autonomous mechanisms.

## 1. Introduction

Cancer represents the second most common cause of death in developed countries and the number of cancer-related deaths is expected to grow due to the increase in life expectancy and lifestyle risk factors [1]. Although surgery and pharmacological treatments have improved patients’ survival, the results remain dismal mainly for late-stage cancer [1]. The overwhelming majority of cancer mortality is caused by metastasis, a complex process that remains the least understood aspect of cancer biology [2]. Metastasis is a process in which cancer cells disseminate from the primary tumor and seed new colonies at distant sites. It involves the local invasion of primary-tumor cells into surrounding tissue, intravasation of these cells into the circulatory system, and subsequent extravasation to other tissues through the vascular walls [2]. 

In this way, cancer cells travel to the parenchyma of a distant tissue and seed microscopic colonies that proliferate to form metastatic lesions. In addition to cancer cell-autonomous mechanisms, metastatic growth depends on the interactions of cancer cells with their niche microenvironment and the crosstalk with various stromal cells, including endothelial cells, fibroblasts, and cells from the innate and adaptive immune system [3]. Over the last years, the biological programs that underlie the dissemination and metastatic outgrowth of cancer cells have begun to emerge. An important aspect is the diversification and adaptation of cancer cells that can be achieved by two main biological processes (1) the dormancy programs (DP) characterized by the activation of quiescence and survival pathways and (2) epithelial-mesenchymal transition (EMT) in which epithelial cells lose their cell polarity and cell-cell adhesions, and gain migratory and invasive properties to become under certain conditions mesenchymal cells that sometimes present stem cell-like properties [3,4]. Notably, in addition to phenotypic differences, significant genotypic diversity exists within tumors, a process known as intra-tumor heterogeneity (ITH) that can be observed at the genetic, proteomic, morphological, and environmental level [5]. 

One of the central drivers of intra-tumor diversification is the chromosome copy number, instability of particular loci, large chromosome segments, or entire chromosomes. This instability may alter the chromosomal content of a cell (aneuploidy). Karyotypic heterogeneity in tumor cells derives from “chromosomal instability” (CIN), a hallmark that not only generates abnormal aneuploid karyotypes, but also expands continually phenotypic heterogeneity as tumor cell populations undergo consecutive cell divisions [6]. One of the major advances in our understanding of the effect of CIN during carcinogenesis comes from the observation that complex aneuploidies are features of tumor types with a predilection for metastasis, treatment resistance, and decreased overall survival. Recurrent examples can be found in microsatellite stable colorectal cancers, triple-negative breast cancer, pancreatic and hepatobiliary cancers, lung cancer, anaplastic thyroid cancer, castrate-resistant prostate cancer, poorly differentiated sarcomas, gynecologic tumors with serous histology, and glioblastoma [6,7].

## 2. Aneuploidy and CIN: Two Sides to the Debate in Cancer

The link between chromosomal abnormalities and cancer was first proposed by the German biologist, Theodor Boveri, over a hundred years ago [8]. Decades of studies have shown that errors in mechanisms of cell division are an important source of genomic diversification that promotes ITH and cancer evolution. Paradoxically, despite the observation that aneuploidy is an unfavorable state for cancer cells, most healthy cells such as osteoclasts or hepatocytes are highly aneuploid [8]. Therefore, aneuploidy is increasingly recognized as a factor that might promote genetic diversity. Aneuploidy is present in around 80% of human solid neoplasms [9]. Cancer cells often experience two major forms of aneuploidy, numerical and segmental. The first is caused by errors in chromosome partitioning during mitosis involving entire chromosomes that vary within and among tumor cells [9,10]. The second is defined by copy number variations of subchromosomal regions [9,10]. While aneuploidy can be readily assessed using classical genetic approaches, CIN can only be indirectly inferred to establish the ongoing rate of chromosome missegregation. With few exceptions as is the case for some hematologic malignancies, aneuploidy and CIN are usually linked in human cancer [9,10]. In addition to CIN, tumors present an alternative type of instability named microsatellite instability or single nucleotide instability. This occurs at the level of base substitutions, deletions, and insertions of one or a few nucleotides, providing evidence that substantial heterogeneity in the amount and type of instability exists both within and between cancer types [9,10,11]. While instability at nucleotide level is often caused by defects in DNA repair, the origin of CIN and aneuploidy remains highly controversial and debated [9,10,11,12]. For example, using patient-derived tumor organoids a recent work showed that ongoing CIN is common in colorectal carcinomas regardless of background genetic alterations, including microsatellite instability [13]. In addition, a higher level of CIN promotes adenoma formation in the distal colon but has no effect in the small intestine [13]. Thus, despite various studies supporting that CIN can be potently oncogenic, certain levels of CIN can have contrasting effects in distinct tissues.

Notably, chromosome segregation errors or DNA replication stress can lead or alternatively fuel aneuploidy through polyploidy, a condition in which a cell contains more than two genome copies [14,15]. A type of polyploidy, in which a single cell has four sets of chromosomes, named tetraploidy, is considered a transitory state on the route to aneuploidy and CIN [16,17]. Whole-genome doubling may promote an elevated level of chromosomal loss and gain, facilitating oncogenic transformation and cancer progression through the production of aneuploid cells. Mitotic errors including mitotic slippage or cytokinetic failure, are considered the main routes to tetraploidy from diploid cells [18]. Mitotic slippage is a phenomenon in which mitotic cells enter the next cell cycle without undergoing chromosome segregation and cell division. Often cancer cells acquire resistance to anti-mitotic drugs avoiding subsequent cell death through mitotic slippage [16]. Cytokinesis failure occurs when the cleavage furrow formation or resolution is disturbed, resulting in binucleated cells [16,17,18]. Cytokinesis failure leads to both centrosome amplification and production of tetraploid cells, which may set the stage for the development of tumor cells (Figure 1A,B). However, tetraploid cells are abundant components at the organism and sub-organism levels in normal tissues, including the liver and heart, indicating that polyploid cellular processes are physiologically relevant and required in generating biodiversity and biocomplexity [17]. The reason why tetraploidy is both beneficial and detrimental for cellular fitness, depending on the cellular context, remains unknown. Other errors in chromosome segregation during mitosis that can lead to karyotype heterogeneity include the formation of micronuclei (Figure 1A,B). 

Micronuclei are small structures containing genetic material that contain less DNA repair and replication machinery, and are often prone to rupture, increasing the further accumulation of chromosomal abnormalities [13,14,15]. Chromosomes enclosed in micronuclei are subjected to more DNA damage and can become exposed to the cytoplasm after micronuclear envelope rupture. This mechanism has been proposed to promote massive structural chromosomal rearrangements, known as chromothripsis, the formation of extrachromosomal DNA as well as double minutes, which can be subjected to strong selective pressures and be present in hundreds of copies per cell [18]. More recently, an intriguing aspect of CIN has been identified whereby chromosome segregation errors, as well as replication stress, can activate innate immune signaling through the introduction of genomic double-stranded DNA (dsDNA) into the cytosol and engagement of the cGAS-STING cytosolic dsDNA sensing antiviral pathway [19]. This new dimension adds complexity to understand the role of CIN in tumor evolution and reveals that the consequences of CIN are not only tumor-cell autonomous, but also involve their crosstalk with the immune microenvironment [19,20]. Recent works highlight the functional role of the centrosome at the crossroad between CIN and inflammatory processes.

## 3. Centrosome Biology and CIN

In animal cells, the centrosome is the major microtubule-organizing center (MTOC) [8]. The core of the centrosome consists of centrioles of nine microtubule (MT) triplets embedded in an ordered and protein-rich matrix called pericentriolar material (PCM). Centrosomes promote the production of bipolar mitotic spindles and supply a matrix of primary cilia in various cell types (Figure 2A) [21,22]. In addition to these structural functions, centrosomes and primary cilia have also evolved into essential signaling hubs to build and regulate the sensory/motor characters of metazoans [23,24]. It is increasingly recognized that centrosome is an organizing unit not only for MTs, but also for actin, Golgi apparatus, and signaling molecules coordinating cell migration, cell polarity, and fundamental signal transduction pathways [25,26]. Large-scale proteomic studies have revealed that there are more than 400 centrosome-associated proteins (about 3% of human proteome, but the exact functions of most of these proteins are still unknown and require further investigation. However, recent advances in imaging, proteomics, and structural biology have revealed new insights into how microtubule-associated proteins regulate the structure, length, and stability of centrosome/cilium complex opening up new avenues for future research [8,26,27,28]. 

In cycling cells, centrosome-associated proteins act to mediate enzyme-substrate interactions and coordinate post-translational modifications to integrate centrosome control with other cellular functions such as the duplication of genetic material required for proper cell cycle progression. While centrosome homeostasis is strictly maintained in healthy cells, centrosome aberrations are commonly observed in human tumors. Over a century ago it was already noted that centrosome aberration is a hallmark of cancer cells, indicating its direct participation in cell transformation [8]. Historically, investigations on the relationship between centrosome aberrations and neoplastic growth have primarily focused on the ability of extra centrosomes to promote tumorigenesis. Studies in D. melanogaster fly or mice models have clearly shown that centrosome amplification is sufficient to induce tumor formation [29]. However, the impact of centrosome alterations on cancer development in humans is less clear because of the intrinsic heterogeneity of cells within tumors and the lack of systematic studies on centrosome alterations during cancer development. Despite a lack of understanding to what extent centrosome aberrations are involved in human carcinogenesis, numerous observations suggest that centrosome alterations are observed at all stages of tumor development. 

## 4. Centrosomes, Mitotic Errors and Cellular Motility 

Tumors often present at least two different types of centrosome alterations, numerical and structural. Numerical alterations are an increase in centrosome copy number and may arise from defects in centrosome duplication, or perhaps more often as a consequence of failed cell divisions (Figure 1A,B). In contrast to centrosome amplification, structural centrosome abnormalities in human tumors are less well characterized. They include altered centrosome size, shape or position, and increased centriole length with the most straightforward defects being an increase in centrosome size as a result of an expansion of the PCM [8,28]. Numerical and structural centrosome aberrations often coexist within the same tumor. Structural and numerical centrosome alterations can reorganize the microtubule cytoskeleton and disrupt tissue architecture, potentially providing a platform for metastatic cell dissemination. Centrosome amplification or loss have remarkably similar effects in promoting CIN. Although little is known about the mechanisms of centrosome elimination, recently, it has been reported that centrosome loss occurs in primary prostate tumors, inducing mitotic errors, producing aneuploidy and multinucleated cells [29]. (Figure 1A,B). Loss of the centrosome linker protein rootlein (encoding for *CROCC* gene) has also been observed in unusual colorectal cancers referred to as the rhabdoid phenotype. Mutations in *CROCC* were shown to produce severe CIN and chromosome segregation errors. Perturbations in centrosome splitting, including other caretaker genes, such as adenomatous polyposis coli (APC) may contribute to aggressive cancer subtypes [8,30]. Centrosome aberrations may induce the dissemination of metastatic cells through at least two ways including: (a) the extrusion of damaged cells toward the basal surface of epithelial monolayers through a phenomenon named ‘‘budding’’ and (b) the formation of invasive protrusions (invadopodia) in mammary epithelial [31]. For example, overexpression of the Ninein-like protein (Nlp), a centrosome protein that interacts with the γ-tubulin ring complex, can sensitize the damaged epithelial cells to basal extrusion (Figure 2B). This switch in directionality from apical to basal dissemination includes a profound reorganization of the microtubule cytoskeleton upon induction of cell damage by etoposide [31] and ref. therein. In addition, overexpression of the centrosome protein CEP131 promotes distinct alterations in the centrosome structure. In this case, the basal extrusion of dying cells with CEP131-induced structural centrosome aberrations occurs in the absence of any external damaging agent [31]. If extruded cells present additional alterations in oncogenes that promote survival, it is plausible that a reversal in the directionality of cell extrusion caused by centrosome aberrations could contribute to the dissemination of metastatic cells (Figure 2B). Finally, it is well known that centrosome amplification induced by Plk4 overexpression leads to increased microtubule nucleation and dynamics resulting in the activation of Rac1-mediated signaling and increased actin polymerization. This pathway promotes the formation of invasive protrusions (invadopodia) in cells with extra centrosomes (Figure 2C). These studies suggest that centrosome is increasingly being recognized as a major communication center for signal transduction pathways and as a center for proteolytic activities with profound effects on cellular motility.

## 5. Rho GTPases Signaling and Centrosome Aberrations

Proteins controlling microtubule dynamics and processes that require changes in cell shape and motility are important for tumor dissemination. Rho GTPases (Ras homologous family) comprise the largest subfamily cluster of the Ras-homology superfamily. Rho GTPases exist in an inactive GDP and an active GTP form [32]. There are approximately 70 different RhoGEFs (Rho Guanine nucleotide exchange factor) and 80 different RhoGAPs (Rho GTPase activating proteins), all of which have a unique spectra of affinities for the different Rho GTPases [32]. The active forms of Rho family members bind to numerous effector proteins and appear to be crucial for various biological processes. Rho GTPase signaling is commonly altered in human tumors, and an elevated expression and/or activation of Rho GTPases often correlates with tumor progression, metastasis, and poor prognosis [33]. Rho GTPases, including Rac, Cdc42, and Rho, play a role in the establishment of cell-cell contacts and cell-matrix interactions. They are crucial to attaining a fully polarized epithelial state and are known for their actions regulating actin cytoskeleton and transcriptional activation [32,33]. 

The downstream targets of Rho GTPases include not only adaptor proteins and kinases, which regulate actin cytoskeleton organization, but also transcription factors, leading to the expression of genes necessary for the drastic cellular morphological changes. For example, cell mesenchymal-type migration is characterized by an elongated cellular shape that requires extracellular proteolysis and integrin engagement, which depends on Rac1-mediated cell polarization and lamellipodia formation. In amoeboid motility, where cells have a rounded morphology, the movement is independent of proteases but also requires high Rho GTPase activity to drive elevated levels of actomyosin contractility [32,33]. Current evidence supports that Rho GTPases also regulate local dynamics of microtubules, centrosome activity, and the function of the Golgi/centriole complex [33,34,35]. Centrosome disruption induces excessive Rac1 activation around the cell periphery, causing rapid focal adhesion turnover, a disorganized actin network, randomly protruding lamellipodia, and the loss of cell polarity [35]. This supports that the centrosome integrity guides the spatial activation of Rac1 to control normal cell polarization and directed cell migration (Figure 2D).

Conversely, centrosome amplification in cultured cells also activates Rac1. In fact, in cells with extra centrosomes, increased centrosomal microtubule nucleation leads to Rac1 activation, disruption of cell-cell contacts, and invasive behavior [8,33]. In human non-transformed cells, these responses referred to as mitotic surveillance pathways depend on p53-activating signaling via the PIDDosome (a multiprotein complex that includes p53-induced death domain protein 1, PIDD1). They lead to activation of caspase-2 (CASP2) or components of the Hippo pathway [8]. This hypothesis agrees with the observation that overexpression of Rac1 is frequently detected in cancers with mutant TP53. Although many in vitro and in vivo data indicate that activated Rho GTPases have tumor-promoting effects, tumor-suppressive functions have also been described for Rho. These contrasting effects of Rho GTPases in cancers may be due to cell-type-specific functions or insufficient available cancer models. More recently, identification of point mutants in the Rho GTPases Rac1, RhoA, and Cdc42 in human tumors has finally given rise to a new paradigm, although the functional and clinical significance of many of these mutants remains poorly understood [32,33]. Recently, multi-region whole-exome and RNA sequencing data from different tumor regions have shown that pathways involved in cell proliferation and mitosis, including signaling by Rho GTPases retain uniform expression levels within individual lung tumors [36]. This supports the hypothesis that biomarkers based on Rho GTPases could help to refine the prediction of patient responses to specific therapies, including those manipulating the immune microenvironment. The Chr17q region containing the centrosomal ubiquitin ligase TRIM37 gene has recently been shown to be frequently amplified in neuroblastoma and breast cancer, rendering these cancer types highly sensitive to Serine/Threonine Protein kinase PLK4 (Polo-like Kinase 4) inhibition [37]. Regarding Rho GTPase, yet few compounds targeting its related signaling networks have been developed beyond an early preclinical stage [22]. Due to the challenges of inhibiting Rho GTPase activation directly, targeting these effectors remains a promising but still unproven approach.

## 6. Centrosome, Cell Cycle and Inflammatory Responses 

In spite of its connection to aggressiveness, it remains to be fully understood the precise contribution of chromosomal instability to cancer phenotypes. Recent reports have shown that genomic instability and DNA damage leads to DNA and cGAS–STING-induced inflammation signaling, which affects cellular antigen presentation [9,38]. These pathways are often triggered by non-canonical NF-kB (Nuclear factor-κB) signaling, as well as coopting myeloid cell mobility programs. Centrosome aberrations (numerical or structural) can be associated with additional factors such as age, inflammation, hypoxia, other environmental influences, or a combination of circumstances. Cells have evolved mechanisms to generate several inflammatory response systems to tackle DNA and centrosome lesions in order to maintain their genome integrity [39]. For example, pro-inflammatory signals through IKKα (Inhibitor of NF-kB kinase α) activation induce nucleophosmin (NPM) hexamer formation, which in turn, leads to the association of NPM with centrosomes in M phase in the case of human cells or in the M phase and interphase in mouse cells. Consistently, loss of IKKα or NPM, decreases the levels of NPM hexamers and its association with centrosomes, thereby promoting centrosome amplification (Figure 3A) [40]. Although IKKα-NPM axis may suppress tumor progression through maintaining proper centrosome duplication in a pro-inflammatory microenvironment, the underlying molecular mechanisms on the interplay between IKKα-NPM axis and centrosomes deserve future investigations.

Nevertheless, centrosome polarization is required for full activation of T lymphocytes, including the generation and secretion of cytokines highlighting the relevance of centrosome translocation [8]. In fact, centrosome maturation, expansion of PCM that occurs as cells pass through specific phases, is essential for the secretion of a number of cytokines such as IL-6 (Interleukin 6), IL-10, and MCP1 (Monocyte chemoattractant protein-1), but not TNF-α (Tumor necrosis factor-α) [41]. Pro-inflammatory stimuli activate interphase centrosome maturation in both immune and non-immune cells through a mechanism dependent on MLK (mixed-lineage kinase) and p38 or JNK (Figure 3B). More recent observations support a model in which supernumerary centrosomes in cancer cells can promote the overproduction and secretion of cytokines and pro-invasive factors, such as IL-8, ANGPTL4 (Angiopoietin Like 4), and GDF-15 (Growth Differentiation Factor 15). Notably, conditioned media from cells with extra centrosomes induce the formation of invasive protrusions in 3D cell organoid cultures with a normal number of centrosomes independently of Rac1, a phenomenon called non-cell-autonomous extra centrosomes-associated secretory pathway (ECASP) [31]. (Figure 3C). Other convincing evidence for the link between centrosome biology and inflammation comes from the observation that cells from patients that have a mutant pericentrin gene are susceptible to infections and their immune response is defective [41]. It is also remarkable that patients with human hereditary disorders carrying mutant centrosomal genes *ALMS1* (Alstrom Syndrome) and *CEP250* (Retinitis Pigmentosa) show a severe deficit in the immune response, inflammation, and extracellular matrix (ECM)-cell interactions [42,43]. In this context, p38 MAPK kinase plays a key role balancing centrosome dynamics regulation with cell cycle and inflammatory responses, for example, controlling mitotic entry timing [44]. Notably, loss of centrosome integrity activates p38 MAPK leading to a p38-p53-p21-dependent G1-S arrest, highlighting the important role played by p38 in maintaining chromosome stability and an attenuated inflammatory response [45] (Figure 3D). The centrosome defects that activate p38 have also been involved in the induction of cellular senescence. This is an irreversible type of growth arrest that is produced in cleavable cells that suffer extensive intrinsic and/or extrinsic damage which connects aging and cancer affecting also immunity [44,45].

In cancer cells, loss of p38 allows aneuploidy tolerance by increasing HIF-1α and glycolysis, limiting metabolic collapse [46]. There is also evidence indicating that MLK3 (Mixed-lineage protein kinase 3), p38, and MK2 (also named MAPKAPK2 (MAPK-activated protein kinases)) are associated with centrosome alterations and DNA replication stress through R-loops [47]. Moreover, p38 MAP-kinase pathway is a known negative regulator of interferon signaling downstream of STING (Stimulator of interferon genes). Hence, at late stages of viral infection, p38-mediated phosphorylation of USP21 (Ubiquitin Specific Peptidase 21), a deubiquitinating enzyme, inhibits STING. p38 pathway appears to be also active in tumor cells with unstable chromosomes in response to the stress induced by chromosome missegregation and endogenous DNA damage [9]. Interestingly, pharmacologic inhibition of p38, selectively regulates type I interferon signaling downstream of STING, whereas other STING-related pathways are not affected (Figure 3D). Therefore, the effects of cGAS-STING activation in cancer depend on the context, being mainly affected by the ongoing aneuploidy state and the level of activity of p38 MAPK. 

## 7. p38MAPK as a Key Mediator of Chromosome Stability and Cell Cycle

The maintenance of chromosome number and stability is a crucial task for cells and involves hundreds of genes with functions in DNA repair, replication, recombination, chromosome segregation, and cell cycle control, among others [48]. Genetic analyses have shown that mutations in the genes encoding p53 or other crucial G2/M-phase checkpoint proteins such as p38MAPK, as well as those involved in mitotic regulation (*ATM*, *CHK2*, *SECURIN*), result in genomic instability [49,50]. The precise role of these genes on CIN is not currently clear, but they are involved in proper chromosome segregation and the activation of cell cycle checkpoints. One of the main causes of CIN is the deregulation of the cell cycle [49]. p38MAPK regulates cell cycle in different situations, controlling genomic instability. For example, p38MAPK controls cell cycle at G0, G1/S, and G2/M transitions to ensure genetic integrity and stability of the cell at each step [49]. p38MAPK also regulates actin cytoskeleton organization, which is required for cytokinesis and mitosis [50,51]. Hence, p38α MAPK deficiency in hepatocytes induces actin disassembly and cytokinesis failure, which leads to the generation of genetically unstable polyploid cells [51]. Furthermore, inhibition of p38MAPK in combination with taxanes increases genomic instability and DNA damage, impairing DNA replication in breast cancer cells [52]. Some mitotic errors can lead to CIN, chromosome mis-segregation, and aneuploidy [53]. p53 has been proved to regulate CIN surveillance and chromosome segregation [54]. Thus, chromosomal mis-segregation frequently leads to activation of p53, which in turn arrests cell cycle, inducing senescence or apoptosis [53]. Several observations indicate that p38MAPK, a kinase activated by various types of stress [55,56] also plays a pivotal role in chromosome mis-segregation and aneuploidy tolerance upstream of p53 [57,58,59] (Figure 3D). This is supported by studies showing that pharmacological chemical inhibition of p38MAPK overcomes p53-dependent cell-cycle arrest after prolonged mitosis or chromosome mis-segregation [60,61] and enhances CIN [62]. It is known that activation of p38MAPK in response to DNA damage induces a G2/M cell cycle checkpoint to repair DNA through p53-dependent mechanisms [60,61,62] (Figure 3D). Although less known, p38MAPK is also able to mediate cell cycle arrest in response to CIN. The most accepted mechanism involves the activation of p38α MAPK and p53 at the centrosomes during spindle assembly in mitosis. The mechanism of arrest involves a prolonged prometaphase that triggers an irreversible p38MAPK-p53 activation followed by a block of cell cycle progression [63]. Posterior studies indicated that p38MAPK activation promotes p53 stability and p53 dependent-apoptosis by suppressing HIF-1α (Hypoxia-inducible factor-1α) among other mechanisms [64]. In fact, p38MAPK activity is not only a sensor of mitosis malfunctioning, but also a positive effector of proper spindle assembly and controls mitotic entry by phosphorylation of Cdc25B (Cell division cycle 25 homolog B) in normal cells [65,66]. In this line, phosphorylated p38MAPK and MK2 colocalized with Plk1 in the spindle poles during prophase and metaphase of normal cells [67]. In addition, functional analysis in mouse oocytes indicates that p38MAPK acting through MK2 regulates spindle assembly and kinetochore-microtubule attachment of chromosomes for accurate chromosome segregation [67]. Disruption of the cytoskeleton will send a stop signal, leading to p53 activation by p38MAPK and cell cycle arrest to allow appropriate distribution. Furthermore, monastrol treatment which inhibits the mitotic kinesin, Eg5, leads to chromosome mis-segregation and causes a p38MAPK–dependent cell cycle delay response accompanied by nuclear accumulation of p53 and the cyclin kinase inhibitor, p21 [68]. Several mechanisms have been proposed to explain how mis-segregated chromosomes activate p38MAPK, including denatured protein accumulation, mechanical stress, or DNA damage [34,50]. Centrosome abnormalities are additional causes of CIN and p38MAPK has emerged as a kinase with a key role in centrosome dysfunction. Hence, the inhibition of p38MAPK rescued cell cycle progression after depletion of centrosome proteins [69]. Evidence also shows that p38MAPK activity is essential for centrosome normal functioning. For example, p38MAPK localizes at the mitotic centrosome allowing chromosomal segregation [69]. Furthermore, localization of p38α MAPK at the kinetochores and the centrosome is also essential for proper chromosomal segregation [68,69]. Active p38MAPK has also been localized at other structures such as the centriolar satellites, discovering a p38MAPK/MK2/14-3-3 signaling cascade that targets centrosome functions and modulates its response to cell stress [69]. All this evidence point to p38MAPK as a key component of a feedback pathway quality control that operates in mitosis and cell division to detect and transform chromosome alterations into a robust G1 arrest, often dependent on p53. 

## 8. p38MAPKs in Aneuploidy, Inflammation and Immune Evasion

Chronic inflammation promotes tumorigenesis and cancer progression, which is associated with an increase in cancer cell survival, invasion, and angiogenesis [70]. Among the different signaling pathways controlling inflammation, it is important to underline the relevant role of p38 MAPKs regulating the expression of pro-inflammatory molecules. p38α MAPK is involved in the induction of the expression of several inflammatory cytokines [70] such as TNF-α, IL-1, and IL-6 and other mediators of inflammation such as cyclooxygenase 2 (COX2), contributing to the development and progression of gliomas, breast cancer, head and neck squamous cell carcinoma, skin cancer or colorectal cancer (CRC) [71,72,73,74]. The regulation of these pro-inflammatory molecules by p38α pathway includes transcriptional and posttranscriptional mechanisms [74,75,76,77]. MK2, a p38α downstream kinase, plays a key role in this posttranscriptional regulation by stabilizing mRNAs and promoting translation [73]. For example, MK2 increases interleukin IL-6 expression through stabilization of its mRNA, while TNF-α production is enhanced by promoting its translation [75]. RNA binding proteins (RBDs), such as AUF-1, HuR (Human antigen R), and TTP (tristetraprolin) interact with mRNAs AU-rich sequences in the 3’Unstranslated region (UTR) to regulate their stability and MK2 controls the activity of these proteins [75]. In particular, TTP promotes mRNAs degradation, action inhibited by p38 and/or MK2 [78,79,80], for example, to increase TNF-α mRNA stability [81,82] and other mRNAs involved in inflammation and cancer growth such as COX-2 (Cyclooxygenase-2), VEGF (Vascular endothelial growth factor), and IL-10 [3]. 

A pro-inflammatory role for p38γ/δ has also been assigned in colon cancer [82,83,84] and skin cancer [85,86]. However, the precise function of p38-dependent inflammation and immune response in tumor development is complex and it might depend on tumor type, stage, and microenvironment. Hence, for example, p38α deficiency in mice favors the development of hepatocellular carcinoma (HCC) associated with inflammation [87], which suggests that p38α is a negative regulator of inflammation-dependent transformation. The function of different p38 isoforms has been particularly studied in CRC. For example, p38α plays a dual role in CRC, having opposing functions in different stages of the disease [88]. In the initial stages of azoxymethane/dextran sodium sulfate (AOM/DSS) induced colitis-associated CRC, p38α expressed in epithelial cells prevents inflammation-associated epithelial damage by decreasing immune cell infiltration and inhibits tumorigenesis. In contrast, it induces proliferation and survival of tumor cells. Hence, p38α inhibition or down-regulation decreases CRC tumor growth [88,89,90], being relevant for it the decrease in the expression of the pro-inflammatory cytokine IL6 [91]. Moreover, MK2-mediated increase in TNF-α, IL-6, and IL1β promotes AOM/DSS-induced colitis-associated CRC and tumor growth in a syngeneic CRC xenograft model [89,90]. In addition, p38 nuclear translocation plays a relevant role in colitis-associated CRC [92]. The deletion of p38α in different cell types has allowed the identification of specific functions for p38α in various cell types during CRC development and progression, highlighting its role in the inflammation mediated by immune cells. Hence, the deletion of p38α in dendritic cells protects from chronic colitis-associated CRC by suppressing the inflammatory response through increasing IL27 levels [93]. Myeloid p38α also contributes to colon inflammation and tumorigenesis in colitis-associated CRC by generating IGF1 [94]. p38γ/δ also play a pro-inflammatory role in colitis-associated colon cancer [83,84,85]. Consequently, p38γ and p38δ deletion decreased inflammation and tumor formation, which is associated with a reduction in cytokines that depends on the hematopoietic cell response [83,84,85]. Cancer immune evasion is a major obstacle in the design of effective anticancer therapeutic strategies. The crosstalk between tumors and the host immune system can both prevent and promote tumor growth and has been identified as a hallmark of cancer [95]. In general, tumor cells are identified and eliminated by the innate and adaptive immune systems [96]. However, if tumor cells are not completely removed, they might enter a dormant state that is reversible that can last many years, during which, they can evade immune surveillance [97]. Finally, tumor cells may escape from the control of the immune system and proliferate in an unrestricted way, leading to new clinically apparent tumors [98]. Tumor cells that escape from the host immune system attack use different strategies such as selection of tumor variants resistant to immune effectors and generation of an immune suppressive environment within the tumor, among others [99]. p38 MAPK can regulate the immunosuppressive properties of some immune cells, such as Myeloid-derived suppressor cells (MDSCs), natural killer cells, or dendritic cells (DCs) [93,94,100,101]. For instance, p38MAPK can promote MDSCs migration to tumors facilitating tumor immune evasion. It has been shown that transmembrane TNF-α induces the expression of CXCR4 by MDSCs via TNFR2 through activation of both NF-κB and p38 MAPK pathways [101]. This CXCR4 upregulation induces chemotaxis of MDSCs to tumors and facilitates evasion of immune surveillance [101,102]. In addition, CXCR4-induced migration of myeloid BMDCs through p38 MAPK activation favors metastasis formation [102]. However, recently it has been shown that LPS treatment induced the expression of iNOS (an M1 marker) in Gr-1^+^CD115^+^ monocytes through p38 MAPK activation, which decreased their suppressive role on CD4 T cells [103,104]. These results suggest that LPS-mediated inflammation via p38MAPKs can inhibit tumor growth by altering the immunosuppressive microenvironment and polarizing M2 Gr-1^+^CD115^+^ monocytes to M1 [103]. p38 MAPKs also regulates dendritic cells (DC) functions [104,105,106,107], promoting immune tolerance. For instance, in a melanoma model that upregulates RET, where it suppresses antitumor T-cell immune responses and promotes melanoma growth [105]. Cisplatin also promotes immune-suppressive tolerogenic DCs by producing IL-10 through a mechanism mediated by p38 MAPK and NF-κB signaling pathways [107]. In agreement with this, the inhibition of STAT3 (Signal Transducer and Activator of Transcription 3) and p38 pathways promote DC differentiation in the tumor microenvironment and increased allogeneic T-cell reactivity against glioma, melanoma cells [106], or multiple myeloma [104]. Furthermore, p38 MAPK inhibition during DC differentiation decreased PPARγ (Peroxisome proliferator-activated receptor γ) expression, which prevents inhibition of p50 transcriptional activities, leading to overexpression of OX40L on DC membrane [100]. This increases DCs ability to activate tumor-specific effector T cells (Teff), blocks Treg conversion and function, and inhibits tumor growth [100]. In agreement with this, it was shown that inducible Treg (iTreg) immunosuppressive activity is maintained by p38 MAPK-mediated IL-10 production [108]. Hence, these reports suggest that p38 inhibition reverts DCs immune-suppressive functions. However, inhibition of p38 only reverts the immunotolerance of CD25^−^ iTreg in the absence of CD25^+^Treg, suggesting that in order to prevent tumor growth and block the immunotolerance of CD25^−^iTreg, it is necessary to deplete CD25^+^Treg, in addition to inhibit p38MAPK [108]. Natural killer (NK) cells and T lymphocytes can generate cytotoxicity through their binding to the major histocompatibility complex class I chain-related molecule A (MHCA) ligands [109]. Interestingly, in pituitary adenoma, p38 MAPKs regulate MICA and MMP-9 expression [110]. Upregulation of MMP9 through p38 MAPK induces MICA cleavage and production of soluble MICA (sMICA), which decreases NK cell cytotoxicity favoring tumor immune escape [110]. Furthermore, deficient expression of immunomodulatory molecules and major histocompatibility complex, among others, can also be regulated by p38 MAPK. For instance, multiple myeloma (MM) cells express a functional TLR, specifically TLR3, that regulates immune surveillance escape [111]. TLR3 activates p38 MAPK pathway, promoting both IFN-γ secretion and IFN-γ-induced cell death [101]. However, in the absence of p38 MAPK activation, TLR3 activates NF-κB that promotes MM cells survival and/or proliferation (Figure 3D) [112]. In addition, p38 MAPK signaling is required for TGF-β1 induced switch of CD8+ T cells to CD8+ Tregs in ovarian cancer microenvironment [113] and other contexts [114]. Activation of p38 MAPK also regulates iTregs anergy through upregulation of the cell-cycle inhibitor p27 [115]. Notably, p38MAPK can also induce tumor cell dormancy, one of the mechanisms used by disseminated tumor cells (DTCs) to avoid immune detection. Dormancy allows survival of DTCs and the colonization of a distant organ [116]. Immune-driven dormancy constitutes a category of tumor dormancy, where the cytotoxic activity of the immune system keeps the proliferating micro-metastasis mass constant [117,118]. The balance between the activation of extracellular regulated kinase (ERK1/2) and actip38α/β was the first signaling mechanism connecting reproducibly to DTC dormancy [118,119]. Moreover, activation of p38 MAPKs induces the dormancy-associated transcription factors DEC2/Sharp1, p27Kip1, p21, and NR2F1 (Nuclear Receptor Subfamily 2 Group F Member 1) that control DTCs reversible quiescence and survival [118,120]. These studies suggest that p38 MAPK axis leads to an increase in cellular heterogeneity, also known as non-genetic heterogeneity, which could generate resistance or persistence to anticancer drugs in patients. 

## 9. Conclusions and Future Perspective

Studies are beginning to shed light on how reprogramming cell cytoskeleton and centrosomes elements can regulate inflammatory signals that are required to maintain chromosomal stability. Chronic inflammation is frequently found surrounding tumors. Notably, cells from patients harboring mutations in centrosome genes often show signs of infection-prone phenotypes, impaired cytokine production, and excess of inflammatory responses, eventually damaging healthy cells, tissues, and organs. The existing evidence suggests that centrosomes play a key role in the regulation of cell senescence, an irreversible type of growth arrest that takes place when cells suffer extensive intrinsic and/or extrinsic damage. Centrosome dysfunction is inseparably linked to aneuploidy and CIN, both hallmarks of tumor cells. Recent advances indicate that centrosome defects (numerical and structural) could contribute to accelerate cancer cell immune evasion through different mechanisms. The inflammatory microenvironment could also result in aneuploidy and spread CIN in tumor cells by inducing a direct genotoxic stress and/or an epithelial–mesenchymal transition (EMT) process. This would lead to a feed-forward loop. Interestingly, the idea that the centrosome can also act as a key coordinator of cellular processes unrelated to microtubule organization, acting for example as a stress sensor, has emerged in recent years. Hence, in addition to well-known cellular stresses such as those induced by DNA damage, oxidative stress, centrosomes abnormalities can also regulate cell cycle arrest and cell senescence through the release of inflammatory mediators. Eukaryotic cell division is a central process that requires complex changes in cytoskeletal organization and function. In recent years, the relevance of Rho GTPases and p38 MAPK in the regulation of many aspects of cell cycle transition, mitosis, and cytokinesis is emerging. Many of these factors and processes have been associated with CIN and inflammation by activating the cGAS-STING pathway. However, the precise nature of these interactions to promote metastasis needs to be fully characterized. Emerging evidence associates CIN to both, promotion and suppression of anti-tumor immunity depending on the type and origin of the tumor. In addition to the well-characterized role of p38 in inflammation and immunity, cell cycle arrest is also regulated by p38, which directly or indirectly influences motor proteins, microtubule dynamics, and centrosome activity. Notably, increasing numbers of evidence points to p53 as a common protein among the different involved pathways. Therefore, in the next future, it will be of great interest to establish the functional significance of centrosomal p53 and p38 activity in relation to various structural centrosome proteins and kinases that regulate them under physiological and pathological conditions. Recent studies on the novel coronavirus SARS-CoV-2, the agent causing the global coronavirus disease 2019 (COVID-19) outbreak have uncovered a viral protein that interacts with human centrosome components. Thus, future investigation on the virus–centrosome interface will help to further understand centrosome biology and designing new drugs against the inflammatory cascade. In addition, future high-resolution and genomic studies will also be necessary to identify how centrosome structural aberrations and their related cellular signaling pathways influence senescence and evasion of immune surveillance. Understanding the molecular mechanisms of these intricate networks will be crucial for the development of more appropriate therapeutic targets.

## Figures and Tables

**Figure 1 biomolecules-11-00629-f001:**
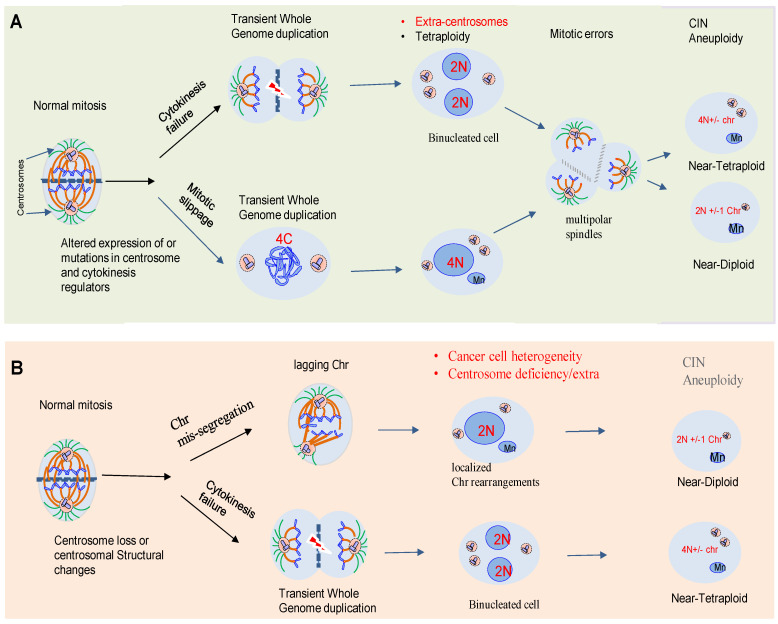
The route to aneuploidy and its link to centrosome dysfunction. (**A**) Aneuploidy is caused by errors in chromosome partitioning during mitosis. Changes include whole chromosomes (numerical aneuploidy) often caused by altered expression or mutations in centrosome (i.e., CEP55) and cytokinesis (i.e., PRC1) regulators [8]. (**B**) Centrosome loss or structural changes in its components are early drivers of genomic instability causing both localized chromosome rearrangements and transient tetraploidy [18]. These alterations will generate intra-tumor heterogeneity and tumors containing a mix of cells with extra-numerical centrosomes or loss of its components. In general, little is known about the mechanisms of centrosome loss. However, centrosomes are normally inactivated or lost during specific developmental stages in different animals. Abbreviations: Chromosome, Chr; CIN; Chromosome instability.

**Figure 2 biomolecules-11-00629-f002:**
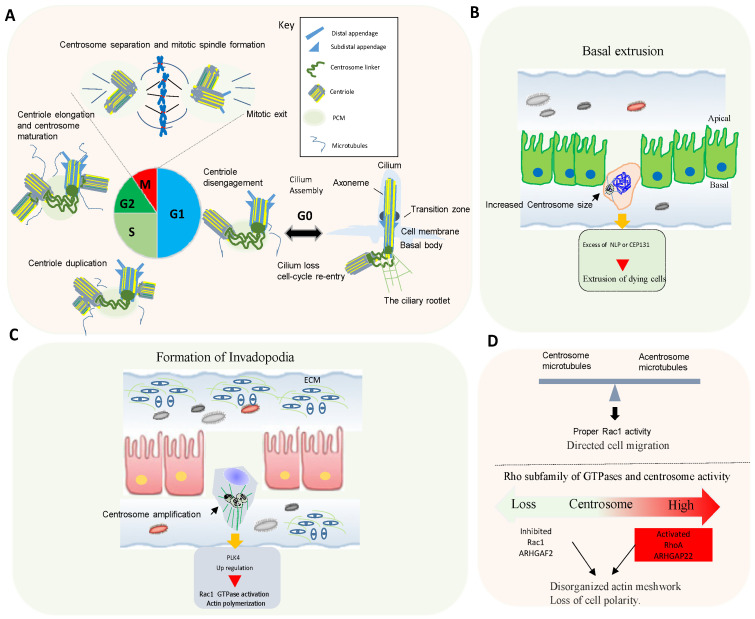
Centrosome cycle and its role in cell motility and invasion. (**A**) In dividing cells, the centrosome duplicates once per cell cycle and segregates in concert with cell-division cycle. The centrosome number and structure are highly regulated during each cell cycle to organize an effective bipolar spindle in the mitotic phase. At the end of mitosis, the daughter centriole disengages from the mother centriole and a centrosome linker is established. When cells enter G0 phase, centrioles can become basal bodies that organize the primary cilium. Shared pathways ensure the coordination between centrosome dynamics, chromosome replication–segregation cycles, and ciliogenesis. Details on the centrosome/cilia dynamins can be found in recent excellent reviews [8,21,23,26,27]. (**B**) Excess of expression of centrosome genes (structural changes) can disrupt apical cell extrusion, instead, causing aberrant basal extrusion. (**C**) Centrosome amplification triggered by overexpression of Plk4 induces the formation of invasive protrusions (invadopodia), which are accompanied by the degradation of ECM components. The increase in centrosomal microtubule nucleation in cells with extra centrosomes promotes activation of the small GTPase Rac1. Rac1 activity, in turn, initiates actin polymerization that disrupt cell-cell adhesion and promotes cell migration. (**D**) Up, the centrosome acts as a controller and balances the formation of centrosomal and acentrosomal microtubules. The presence of centrosome regulates proper Rac1 activity and allows directed cell migration. Down, centrosome activity (loss or activation) regulates differently members of Rho family of GTPases family. Interference or an excess with the formation of centrosome increases acentrosome, microtubules assembly, and activation of Rac1, which in turn leads to the loss of cell polarity.

**Figure 3 biomolecules-11-00629-f003:**
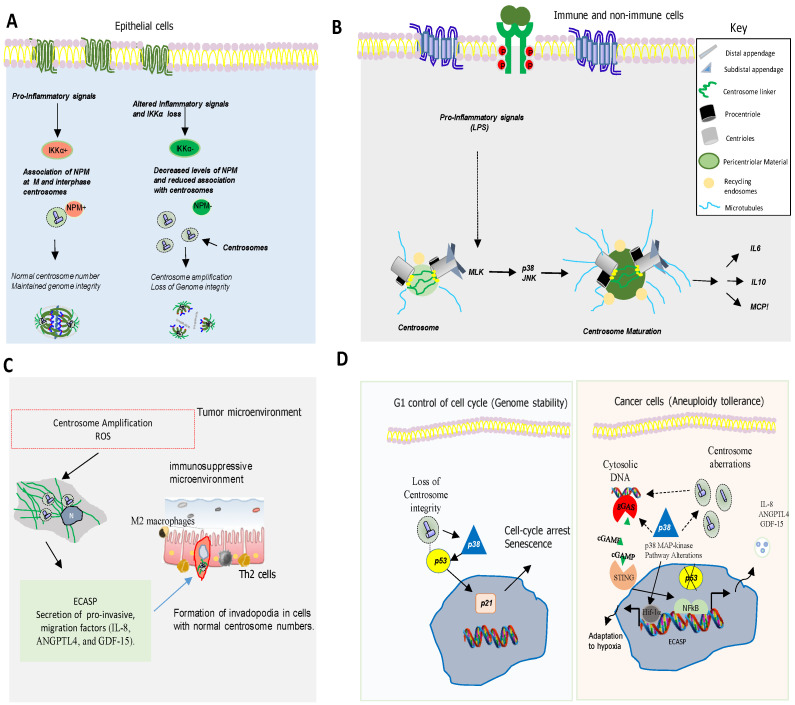
Chromosome instability in cancer and inflammatory responses. (**A**) Pro-inflammatory cytokines activate the IKKα/Nucleophosmin (NPM) hexamer formation and centrosome interplay in mouse and human cells [40]. IKKα activation and increased NPM levels (red) allow NPM-Centrosome interaction, which allows the maintenance of normal centrosome number and genome integrity. Right, IKKα or NPM loss (green), decreases the levels of NPM hexamers and the association of NPM with centrosomes, thereby promoting centrosome amplification and genome changes that predispose to cancer [40]. (**B**) Exposure to LPS induces the axis MLK-p38-JNK which is critical for interphase centrosome maturation and secretion of inflammatory cytokines by immune and non-immune cells. (**C**) Centrosome amplification facilitates the release of proinvasive factors (ECASP) to induce non-cell-autonomous invasion. Although it remains undetermined how the additional centrosomes promote the ECASP, it is dependent on elevated levels of reactive oxygen species in cells with amplified centrosomes. The release of proinvasive factors is required for invadopodia formation in cells with normal centrosomes. Chronic NF-κB activation mediates the ECASP through the regulation of proteins, including IL-8, to shape the immunosuppressive microenvironment. (**D**) Loss of centrosome integrity activates p38-p53-p21 pathway resulting in cell-cycle arrest or senescence acting as a cell-cycle checkpoint pathway. In cancer cells, centrosome dysfunction leads to the generation of cytosolic dsDNA, which in turn, activates the cGAS–STING and alternative inflammatory pathways such as NF-κB signaling leading to ECASP. Centrosome abnormalities in cooperation with p38 and p53 dysfunction can also lead aneuploidy tolerance and adaptation to hypoxia.

## Data Availability

Not applicable.

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
