# Peer review of "Centrosome Dynamics and Its Role in Inflammatory Response and Metastatic Process"

_biomolecules, 2021, doi:10.3390/biom11050629_

Round 1

Reviewer 1 Report

I have read the manuscript of Pancione et al., “Centrosome dynamics and its role in inflammatory response and metastatic process”, to my great benefit.

The authors presented a large body of modern biochemical data on the role of centrosomal proteins in the processes of inflammation and oncogenesis with accuracy and precision in every detail. Unfortunately, the morphological data given by the authors do not presented with the same accuracy. I set out my comments below.

I can recommend to publish this review after correcting the mistakes.

Major points:

Figure 2A

I have serious claims on the presentation in the review of the structural rearrangements of the centrosome in the cell cycle, in particular, shown in Figure 2a. In its current form, this figure gives a completely wrong idea of this process.

In fig. 2A, there are inaccuracies and gross errors that need to be completely corrected.

1) In mitosis, there are no subdistal appendages on the centrioles. You can read in detail about the distal and subdistal appendages in a specially dedicated article, which is freely available (Uzbekov, Alieva, 2018). Who are you, subdistal appendages of centriole? Open Biol. 8: 180062. https://royalsocietypublishing.org/doi/10.1098/rsob.180062

2) The cartwheel structure is typical for vertebrate procentrioles and is absent in the lumen of the daughter centriole in the G1 phase of the cell cycle.

3) Visually, light gray centrioles on a light blue background are poorly visible.

4) In the Figure, the procentrioles in mitosis are shorter than in the G2 phase of the cell cycle, which is obviously incorrect.

5) In mitosis, in fact, in the lumen of none of the 4 centrioles there is no cartwheel structure.

6) Cartwheel structure is the basis for the formation of procentriole, and in the Figure procentrioles are not connected with them in any way.

7) Procentrioles appear on two centrioles already at the end of the G1 phase of the cell cycle, at the "restriction point", when the cell "decides" to go to the next cell cycle.

In general, to correct this Figure, I strongly recommend using the scheme of the correlation of the centriolar and cell cycles available in one of the reviews on this topic in the open access in the journal Cells (Uzbekov, Avidor-Reiss, 2020). Principal Postulates of Centrosomal Biology. Version 2020. Cells 2020, 9(10), 2156; https://doi.org/10.3390/cells9102156

Lines 152-154

In the text the authors wrote "Centrosomes promote the production of bipolar mitotic spindles and supply a matrix of primary cilia in various cell types (Figure 2A)". However, Figure 2A does not depict the second mentioned centrosome function (formation of the primary cilium).

This is a separate big topic, modern look at the role of the primary cilium you can find in the review (Conkar and Firat-Karalar, 2018). https://www.novapublishers.com/wp-content/uploads/2018/09/Flagella-and-Cilia-CH-6.pdf

Line 157

The role of the centrosome in the organization of actin microfilaments in the cell was carried out in (Inoue et al., 2019), which should be cited here.

«It is increasingly recognized that centrosome is an organizing unit not only for MTs, but also for actin  (Inoue et al., 2019).»

Actin filaments regulate microtubule growth at the centrosome. https://www.ncbi.nlm.nih.gov/pmc/articles/PMC6545561/

Lines 313-314, Figure 3A, 3B

«IKKα-NPM axis may use the inflammatory signal to suppress centrosome amplification, promote genomic integrity and prevent tumor progression (Figure 3A).»

The Figure shows centrioles with large procentrioles, that is, the cell has already entered the S phase of the cell cycle. Is it really? Does the cell cycle stop before the start of the procentrioles appearance? Clarify please.

Author Response

Response to Reviewer Comments

Reviewer 1

Point 1: I have serious claims on the presentation in the review of the structural rearrangements of the centrosome in the cell cycle, in particular, shown in Figure 2a. In its current form, this figure gives a completely wrong idea of this process. In fig. 2A, there are inaccuracies and gross errors that need to be completely corrected.

Response 1: We have fully changed the Figure 2A, taking into account all raised recommendations on the structural rearrangements of the centrosome.

Point 2: In mitosis, there are no subdistal appendages on the centrioles. You can read in detail about the distal and subdistal appendages in a specially dedicated article, which is freely available (Uzbekov, Alieva, 2018). Who are you, subdistal appendages of centriole? Open Biol. 8: 180062.

https://royalsocietypublishing.org/doi/10.1098/rsob.180062

Response 2: Subdistal appendages on the mitotic centrioles have been removed accordingly. In addition, the article by (Uzbekov, Alieva, 2018) is now referenced in the manuscript (Ref. 24).

Point 3: The cartwheel structure is typical for vertebrate procentrioles and is absent in the lumen of the daughter centriole in the G1 phase of the cell cycle.

Response 3: The new Figure 2A shows that the cartwheel appears in the lumen of daughter centriole during S phase, but it is absent in G1 phase.

Point 4: Visually, light gray centrioles on a light blue background are poorly visible.

Response 4: We have changed the colours to increase the contrast of centrosome structure. Centrioles now appear in blue/yellow on a light peach background.

Point 5: In the Figure, the procentrioles in mitosis are shorter than in the G2 phase of the cell cycle, which is obviously incorrect.

Response 5: The dimensions of centrioles have been modified and now there are equal in G2 and M phase.

Point 6:  In mitosis, in fact, in the lumen of none of the 4 centrioles there is no cartwheel structure.

Response 6: The cartwheel structure in the mentioned centrioles has been removed.

Point 7: Cartwheel structure is the basis for the formation of procentriole, and in the Figure procentrioles are not connected with them in any way.

Response 7: It is now shown that the cartwheel is closely associated with the formation of procentriole during centrosome duplication.

Point 8: Procentrioles appear on two centrioles already at the end of the G1 phase of the cell cycle, at the "restriction point", when the cell "decides" to go to the next cell cycle.

Response 8: Owing to space constrains, these structural details have not been included. However, we have changed the image to show that centriole duplication starts at the end of the G1 phase and procentrioles are more clearly visible during S phase of the cell cycle.

Point 9: In general, to correct this Figure, I strongly recommend using the scheme of the correlation of the centriolar and cell cycles available in one of the reviews on this topic in the open access in the journal Cells (Uzbekov, Avidor-Reiss, 2020). Principal Postulates of Centrosomal Biology. Version 2020. Cells 2020, 9(10), 2156; https://doi.org/10.3390/cells9102156

Response 9: We have greatly appreciated the recommendation to use the scheme of the paper indicated by you. We cite the mentioned review in the revised version of our manuscript (Ref. 26). This work has also been cited not only in Figure 2A legend, but also in lines 162-170 from the manuscript.

Point 10: Lines 152-154, In the text the authors wrote "Centrosomes promote the production of bipolar mitotic spindles and supply a matrix of primary cilia in various cell types (Figure 2A)". However, Figure 2A does not depict the second mentioned centrosome function (formation of the primary cilium).

Response 10: A brief description of the primary cilia assembly/disassembly process in relation to centrosome cycle is now shown in the new Figure 2A and further discussed in lines 158-164 from the manuscript.

Point 11: This is a separate big topic, modern look at the role of the primary cilium you can find in the review (Conkar and Firat-Karalar, 2018). https://www.novapublishers.com/wp-content/uploads/2018/09/Flagella-and-Cilia-CH-6.pdf

Response 11: The mentioned review is cited in the revised version of the manuscript (Ref.27), in Figure 2A legend and briefly discussed in lines 164-170 from the manuscript.

Point 12: Line 157. The role of the centrosome in the organization of actin microfilaments in the cell was carried out in (Inoue et al., 2019), which should be cited here. «It is increasingly recognized that centrosome is an organizing unit not only for MTs, but also for actin (Inoue et al., 2019).» Actin filaments regulate microtubule growth at the centrosome. https://www.ncbi.nlm.nih.gov/pmc/articles/PMC6545561/

Response 12: The mentioned work has been cited in the revised version of the manuscript (Ref. 25).

Point 13: Lines 313-314, Figure 3A, 3B «IKKα-NPM axis may use the inflammatory signal to suppress centrosome amplification, promote genomic integrity and prevent tumor progression (Figure 3A). The Figure shows centrioles with large procentrioles, that is, the cell has already entered the S phase of the cell cycle. Is it really? Does the cell cycle stop before the start of the procentrioles appearance? Clarify please.

Response 13: To clarify this point, we have modified the manuscript (lines 323-334) as follows: “cells have evolved mechanisms to generate several inflammatory response systems to tackle DNA and centrosome lesions in order to maintain their genome integrity. For example, pro-inflammatory signals through IKKa activation induces nucleophosmin (NPM) hexamer formation, which in turn, leads to the association of NPM with centrosomes in M phase in the case of human cells or in the M phase and interphase in mouse cells. Consistently, loss of IKKα or NPM, decreases the levels of NPM hexamers and its association with centrosomes, thereby promoting centrosome amplification (Figure 3A). Although IKKα-NPM axis may suppress tumor progression through maintaining proper centrosome duplication in a pro-inflammatory microenvironment, the underlying molecular mechanisms on the interplay between IKKα-NPM axis and centrosomes deserve future investigations”. In addition, we have also corrected the Figure 3A and modified the corresponding legend.  

Reviewer 2 Report

The present manuscript is well-structured, well-written and easy to understand. A fly in the ointment, most latest related publications not cited , I strong suggest the author cited those papers.

Cell Cycle-Dependent Dynamics of the Golgi-Centrosome Association in Motile Cells.

Frye K, Renda F, Fomicheva M, Zhu X, Gong L, Khodjakov A, Kaverina I.Cells. 2020 Apr 25;9(5):1069. doi: 10.3390/cells9051069.PMID: 32344866 Free PMC article.

A Proximity Mapping Journey into the Biology of the Mammalian Centrosome/Cilium Complex.

Arslanhan MD, Gulensoy D, Firat-Karalar EN.Cells. 2020 Jun 3;9(6):1390. doi: 10.3390/cells9061390.

DNA Replication Stress and Chromosomal Instability: Dangerous Liaisons.

Wilhelm T, Said M, Naim V.Genes (Basel). 2020 Jun 10;11(6):642. doi: 10.3390/genes11060642.

Ongoing chromosomal instability and karyotype evolution in human colorectal cancer organoids.

Bolhaqueiro ACF, Ponsioen B, Bakker B, Klaasen SJ, Kucukkose E, van Jaarsveld RH, Vivié J, Verlaan-Klink I, Hami N, Spierings DCJ, Sasaki N, Dutta D, Boj SF, Vries RGJ, Lansdorp PM, van de Wetering M, van Oudenaarden A, Clevers H, Kranenburg O, Foijer F, Snippert HJG, Kops GJPL.Nat Genet. 2019 May;51(5):824-834. doi: 10.1038/s41588-019-0399-6. Epub 2019 Apr 29.

Mitotic slippage: an old tale with a new twist.

Sinha D, Duijf PHG, Khanna KK.Cell Cycle. 2019 Jan;18(1):7-15. doi: 10.1080/15384101.2018.1559557. Epub 2019 Jan 2.PMID: 30601084 Free PMC article. Review.

The tumor cells that acquire resistance to anti-mitotic drugs have chromosomal instability (CIN) and are often aneuploid. In this review, we outline the key mechanisms involved in dictating the cell fate during perturbed mitosis and how these processes impede …

Degree and site of chromosomal instability define its oncogenic potential.

Hoevenaar WHM, Janssen A, Quirindongo AI, Ma H, Klaasen SJ, Teixeira A, van Gerwen B, Lansu N, Morsink FHM, Offerhaus GJA, Medema RH, Kops GJPL, Jelluma N.Nat Commun. 2020 Mar 20;11(1):1501. doi: 10.1038/s41467-020-15279-9.PMID: 32198375 Free PMC article.

Most human cancers are aneuploid, due to a chromosomal instability (CIN) phenotype. Despite being hallmarks of cancer, however, the roles of CIN and aneuploidy in tumor formation have not unequivocally emerged from animal studies and are thus still unc …

Chromosomal Instability in Tumor Initiation and Development.

Bach DH, Zhang W, Sood AK.Cancer Res. 2019 Aug 15;79(16):3995-4002. doi: 10.1158/0008-5472.CAN-18-3235. Epub 2019 Jul 26.

Author Response

Reviewer 2

Point 1: A fly in the ointment, mos latest related publications not cited I strong suggest the author cited those papers.

Response 1: We have taken into account your suggestion and we cite the mentioned publications in the revised version of the manuscript (References 12-16 and 21-23).

Round 2

Reviewer 1 Report

The authors responded adequately and meticulously to all the comments I made. I have no more comments on the presented paper and I can recommend it for publication in the journal Biomolecules.